# Immunotherapy in Small Cell Lung Cancer

**DOI:** 10.3390/cancers12092522

**Published:** 2020-09-04

**Authors:** Giovanna Esposito, Giuliano Palumbo, Guido Carillio, Anna Manzo, Agnese Montanino, Vincenzo Sforza, Raffaele Costanzo, Claudia Sandomenico, Carmine La Manna, Nicola Martucci, Antonello La Rocca, Giuseppe De Luca, Maria Carmela Piccirillo, Rossella De Cecio, Gerardo Botti, Giuseppe Totaro, Paolo Muto, Carmine Picone, Nicola Normanno, Alessandro Morabito

**Affiliations:** 1Thoracic Medical Oncology, Istituto Nazionale Tumori, IRCCS “Fondazione G. Pascale”, 80131 Napoli, Italy; espositogiovanna87@gmail.com (G.E.); giuliano.palumbo@yahoo.it (G.P.); anna.manzo@istitutotumori.na.it (A.M.); a.montanino@istitutotumori.na.it (A.M.); v.sforza@istitutotumori.na.it (V.S.); r.costanzo@istitutotumori.na.it (R.C.); c.sandomenico@istitutotumori.na.it (C.S.); 2Department of Oncology and Hematology, Azienda Ospedaliera Pugliese—Ciaccio, 88100 Catanzaro, Italy; guidocarillio@gmail.com; 3Thoracic Surgery, Istituto Nazionale Tumori, “Fondazione G.Pascale”—IRCCS, 80131 Napoli, Italy; c.lamanna@istitutotumori.na.it (C.L.M.); n.martucci@istitutotumori.na.it (N.M.); a.larocca@istitutotumori.na.it (A.L.R.); g.deluca@istitutotumori.na.it (G.D.L.); 4Clinical Trials Unit, Istituto Nazionale Tumori, “Fondazione G.Pascale”—IRCCS, 80131 Napoli, Italy; m.piccirillo@istitutotumori.na.it; 5Pathology, Istituto Nazionale Tumori, “Fondazione G.Pascale”—IRCCS, 80131 Napoli, Italy; r.dececio@istitutotumori.na.it; 6Scientific Directorate, Istituto Nazionale Tumori, “Fondazione G. Pascale”—IRCCS, 80131 Naples, Italy; g.botti@istitutotumori.na.it; 7Radiotherapy, Istituto Nazionale Tumori, “Fondazione G. Pascale”—IRCCS, 80131 Naples, Italy; g.totaro@istitutotumori.na.it (G.T.); p.muto@istitutotumori.na.it (P.M.); 8Radiology, Istituto Nazionale Tumori, “Fondazione G.Pascale”—IRCCS, 80131 Napoli, Italy; c.picone@istitutotumori.na.it; 9Cellular Biology and Biotherapy, Istituto Nazionale Tumori, “Fondazione G.Pascale”—IRCCS, 80131 Napoli, Italy; n.normanno@istitutotumori.na.it

**Keywords:** SCLC, ipilimumab, tremelimumab, nivolumab, atezolizumab, pembrolizumab, durvalumab, PDL-1, tumour mutation burden

## Abstract

**Simple Summary:**

Small cell lung cancer (SCLC) accounts for about 15% of lung cancers and it has limited therapeutic options and poor prognosis. There has been no real progress for over 30 years in the treatment of this aggressive tumor type and platinum based chemotherapy represented the cornerstone of therapy. Immune checkpoint inhibitors are the first agents in the last decades to determine an improvement in outcomes of patients with extensive stage (ES) SCLC patients. In the IMpower 133 and CASPIAN studies, the addition of atezolizumab or durvalumab, respectively, to first-line chemotherapy produced a significant improvement in overall survival with an acceptable safety profile in previously untreated patients with ES-SCLC, leading to a new standard of care. This review summarizes the main results observed with checkpoint inhibitors in SCLC, discussing the critical issues related to the use of novel checkpoint inhibitors and the future research with immunotherapy agents in SCLC.

**Abstract:**

Small-cell lung cancer (SCLC) is an aggressive tumor type with limited therapeutic options and poor prognosis. Chemotherapy regimens containing platinum represent the cornerstone of treatment for patients with extensive disease, but there has been no real progress for 30 years. The evidence that SCLC is characterized by a high mutational burden led to the development of immune-checkpoint inhibitors as single agents or in combination with chemotherapy. Randomized phase III trials demonstrated that the combination of atezolizumab (IMpower-133) or durvalumab (CASPIAN) with platinum-etoposide chemotherapy improved overall survival of patients with extensive disease. Instead, the KEYNOTE-604 study demonstrated that the addition of pembrolizumab to chemotherapy failed to significantly improve overall survival, but it prolonged progression-free survival. The safety profile of these combinations was similar with the known safety profiles of all single agents and no new adverse events were observed. Nivolumab and pembrolizumab single agents showed anti-tumor activity and acceptable safety profile in Checkmate 032 and KEYNOTE 028/158 trials, respectively, in patients with SCLC after platinum-based therapy and at least one prior line of therapy. Future challenges are the identification predictive biomarkers of response to immunotherapy in SCLC and the definition of the role of immunotherapy in patients with limited stage SCLC, in combination with radiotherapy or with other biological agents.

## 1. Introduction

Small cell lung cancer (SCLC) is an aggressive tumor type, accounting for about 15% of lung cancers [1]. Traditional tumor staging divides SCLC in two categories, limited stage (LS-SCLC) and extensive stage (ES-SCLC) disease [2]. The first defines a tumor confined in a hemithorax and encompassed in one radiation field, the latter the cases with disease spreading elsewhere. Most patients (about 70%) present with an extensive disease at diagnosis. The combination of etoposide plus platinum (EP) has been the standard first line treatment of ES-SCLC for many years [3], with an objective response rate of 40–70% [4]. Responses to second line chemotherapy (topotecan or irinotecan) are disappointing: in a phase III trial, topotecan was compared to cyclophosphamide, doxorubicin and vincristine (CAV) in 211 patients with recurrent SCLC. Topotecan resulted in improved control of symptoms (*p* < 0.043) and less hematological toxicity (*p* < 0.001), however, no difference in response rate (24.3% vs. 18.3%, *p* = 0.285), median time to progression (13.3 vs. 12.3 weeks, *p* = 0.552) and median survival (25 vs. 24.7 weeks, *p* = 0.795) were observed between patients treated with topotecan and CAV, respectively [5]. Median overall survival (OS) is 10 months in patients with extensive disease and approximately 20 months for patients with LS-SCLC. Recent evidence supports the relevant immunogenicity of SCLC, based on a high T-lymphocytes tumor infiltration, frequent paraneoplastic syndromes substained by autoantibodies production, and high rate of mutational burden, also related to prolonged tobacco exposure [6]. A correlation between tumor mutation burden (TMB) and objective response rate was described in NSCLC patients receiving immune checkpoint inhibitors, although in a large, randomized clinical study (CheckMate 227) a similar relative benefit was observed with nivolumab plus ipilimumab versus chemotherapy regardless of tumor mutational burden [7,8]. All the above evidences suggest a possible role of immunotherapy also in SCLC.

The immune response against cancer cells starts with antigen-presenting cells (APCs) that uptake and process tumor proteins, and subsequently activate CD4+ helper and CD8+ cytotoxic T cells. However, cancer cells develop the ability to evade the immune system by activating immune checkpoint pathways that are devoted to protect healthy tissues from damage induced by the inflammatory responses and to maintain self-tolerance, thus preventing autoimmunity [9]. Cytotoxic T lymphocyte antigen-4 (CTLA-4) receptor, programmed cell death-1 (PD-1) receptor, both expressed on the T cell surface, and the PD-1 ligands, PD-L1/L2, expressed on the APCs and the tumor cell surface, represent potential targets of the novel immuno-checkpoint inhibitors. CTLA-4 regulates immune responses in the early stages of T cell activation [10], especially in the lymph nodes. CTLA-4 has a higher binding affinity as compared with the CD28 receptor for binding to its ligands, CD80/CD86, expressed by APCs, thus resulting in an inhibitory signal to the T cell. PD-1 is another important immune checkpoint receptor inhibiting T cell activation within peripheral tissues during inflammatory response, in a following stage compared with CTLA-4. The binding of PD-1, expressed in T cell infiltrating the tumor, to its main ligand PD-L1 leads to an inhibition of T cell immune response [11]. Inhibition of CTLA-4 by the immune checkpoint inhibitors, such as ipilimumab or tremelimumab, inhibition of PD-1 by pembrolizumab or nivolumab and inhibition of PD-L1 by atezolizumab or durvalumab, lead to antitumor immune system stimulation and favorable results in terms of overall response rate (ORR), progression-free survival (PFS) and OS in several tumors, including melanoma, kidney, bladder, head and neck, breast, cervical, liver, lung cancer and lymphoma. In particular, in patients with advanced NSCLC, nivolumab, atezolizumab and pembrolizumab have been first approved by the U.S. Food and Drug Administration as single agents in the second-line setting of treatment and then they have been approved in the first-line setting: as single agents in PDL-1 positive patients (pembrolizumab and atezolizumab), in combination with chemotherapy in patients with any histology (pembrolizumab) or non-squamous histology (atezolizumab), or in combination with ipilimumab in PDL-1 positive patients or with ipilimumab and two cycles of chemotherapy (nivolumab). Finally, durvalumab has been approved in unresectable, stage III NSCLC whose disease has not progressed following concurrent platinum-based chemotherapy and radiation therapy.

Moreover, multi-immune checkpoint inhibitors association, by exploiting their complementary mechanisms of action, or combination of immunotherapy and chemotherapy, respectively acting indirectly and directly against the tumor, are an interesting area of exploration in SCLC. This review summarizes the main results observed with checkpoint inhibitors in SCLC, discussing the critical issues related to the use of novel checkpoint inhibitors and the future research with immunotherapy agents in SCLC.

## 2. PDL-1/PD-1 Inhibitors in First Line Therapy

Three randomized clinical trials have demonstrated the efficacy of the combination of atezolizumab (IMPOWER 133) or durvalumab (CASPIAN) or pembrolizumab (Keynote-604) with chemotherapy in the first line treatment of patients with ES-SCLC (Table 1). The IMpower 133 was a phase III, double-blind, placebo-controlled trial of atezolizumab in combination with standard carboplatin-etoposide in patients with ES-SCLC [12]. Eligible patients were randomized to receive carboplatin (area under the curve [AUC] of 5 mg per mL per minute, on day 1) and etoposide (100 mg/m^2^, on days one through three of each cycle) with either atezolizumab (at a dose of 1200 mg, on day 1 of each cycle) or placebo for four cycles, followed by either atezolizumab or placebo until disease progression or unacceptable toxicity. Prophylactic cranial irradiation (PCI) was allowed during the maintenance phase, but consolidation thoracic radiation was excluded. Overall, 403 patients were enrolled: median age was 64 years with almost 50% of patients older than 65 years, most of patients had an ECOG performance status (PS) of 1, and roughly 8% patients presented brain metastases (8.5% in experimental and 8.9% in control arm, respectively). The combination of atezolizumab plus chemotherapy significantly prolonged overall survival, the primary endpoint, and progression-free survival, co-primary endpoint. Median OS was 12.3 months for experimental arm versus 10.3 months for placebo arm (HR: 0.70; 95%CI: 0.54–0.91, *p* = 0.007), while PFS was 5.2 months for atezolizumab arm versus 4.3 months for control arm (HR: 0.77; 95%CI: 0.62–0.96, *p* = 0.02). Safety profile was evaluated among patients who received at least one dose of atezolizumab, with a well balanced number of chemotherapy doses between the two cohorts. The most common grade 3 or 4 adverse events related to the trial regimens were neutropenia, anemia, and decreased neutrophil count, while the most common immune-related adverse events were skin rash and hypothyroidism. Level of PDL-1 expression was not used for stratification. A tumor mutational burden (TMB) subgroup analysis was done through blood samples: no correlation with OS was found at the cutoff of 10 or 16 mutations per megabase [12]. At 2019 ESMO, updated OS data were presented: after 22.9 months of median follow-up of, median OS was 12.3 months in the group of patients randomized to atezolizumab and 10.3 months in the placebo group (HR 0.76; 95% CI: 0.60–0.95; *p*  =  0.0154) [13]. The CASPIAN trial was a randomized, open-label, phase III study that compared durvalumab plus EP for 4 cycles or durvalumab-tremelimumab plus EP for 4 cycles versus EP alone for up to six cycles in 805 patients with ES-SCLC [14]. EP consisted of etoposide 80–100 mg/m^2^ (on days 1–3) plus investigator’s choice of either carboplatin AUC 5–6 mg/mL per min or cisplatin 75–80 mg/m^2^ (on day one of each cycle) administered every 21 days up to six cycles. PCI was allowed after chemotherapy. Patients randomized to the immunotherapy arms were treated up to four cycles of EP plus durvalumab 1500 mg with or without tremelimumab 75 mg every three weeks followed by maintenance durvalumab 1500 mg every four weeks. Median age was 63 years, 62% had an ECOG PS of 1 and 11% had brain metastases. In patients randomized to experimental arm, durvalumab was continued as maintenance therapy in non-progressive patients until progression disease. Durvalumab plus chemotherapy significantly improved median OS compared with the chemotherapy arm (13.0 versus 10.3 months; HR 0.73, 95%CI: 0.59–0.91; *p* = 0.0047). Median PFS was 5.1 months (95%CI: 4.7–6.2) with durvalumab plus platinum–etoposide versus 5.4 months (95%CI: 4.8–6.2) with platinum–etoposide alone. A sustained separation of OS curves was confirmed at the updates analysis presented at ASCO 2020 (HR 0.75; 95%CI: 0.62–0.91, *p* = 0.0032), with 22.2% versus 14.4% of patients alive at 24 months [15]. On the contrary, the combination of tremelimumab plus durvalumab and chemotherapy did not improve OS compared with chemotherapy alone: median OS was 10.4 versus 10.5 months in the experimental versus control arm, respectively (HR 0.82; 95%CI 0.68–1.00; *p* = 0.0451, non-statistically significant according to the protocol). Safety findings were consistent with the safety profile of the drugs: grade 3–4 AEs were observed in 70.3%, 62.3% and 62.8% of patients treated with durvalumab and tremelimumab plus EP, durvalumab plus EP and EP, respectively. AEs leading to treatment discontinuation and immune-related AEs were 21.4%, 10.2%, 9.4%, and 36.1%, 20.0% and 2.6% in the three groups, respectively. Keynote-604 was a randomized double-blind, placebo-controlled phase III study trial of pembrolizumab (200 mg, day 1) in combination with EP (etoposide 100 mg/m^2^ on days 1–3, carboplatin AUC 5 on day 1 or cisplatin 75 mg/m^2^ on day one) versus placebo plus EP in patients with ES-SCLC [16]. Overall, 453 patients were randomly assigned to receive either EP plus pembrolizumab or EP plus placebo for four cycles, followed by pembrolizumab/placebo for up to 31 cycles: roughly 74% had PS 1, 41% had liver metastases and 70% were treated with carboplatin. Brain metastases were reported in 14.5% of experimental arm and 9.8% of control arm. The addition of pembrolizumab to EP improved median PFS (4.5 versus 4.3 months; HR 0.75, 95%CI: 0.61–0.91, *p* = 0.0023), but although it prolonged also OS (10.8 versus 9.7 months), the significance threshold was not met (HR: 0.80; 95%CI: 0.64–0.98; *p* = 0.0164).

The OS 24-month rate was 22.5% versus 11.2% with experimental versus control arm, respectively. Subgroup analysis showed a similar effect in all groups analyzed, with the exception of patients with brain metastases. Objective response rate (ORR) was 70.6% (95%CI: 64.2–76.4) in experimental arm and 61.8% (95%CI: 55.1–68.2) in control arm, respectively. The safety profile of pembrolizumab plus chemotherapy was as expected: grade 3–4 adverse events were 76.7% vs. 74.9% and grade 5 AEs were 6.3% vs. 5.4% with experimental versus control arm, respectively. AEs leading to discontinuation of treatment were more frequent with experimental than control arm (14.8% vs. 6.3%). Immune-mediated AEs were reported in 24.7% of patients treated with pembrolizumab plus EP versus 10.3% of EP: hypothyroidism was observed in 10.3% versus 2.2%, hyperthyroidism in 6.7% versus 2.7%, pneumonitis in 4% versus 2.2%, severe skin reaction in 2.2% vs. 0.9% and hepatitis in 1.8% vs. 0% of patients treated in experimental versus control arm, respectively.

## 3. CTLA-4 Inhibitors in First Line Therapy

The combination of the anti-CTLA4 ipilimumab plus carboplatin and paclitaxel was first evaluated in a randomized phase II study in 130 patients with a new diagnosis of ES-SCLC [18]. Eligible patients were randomized to receive paclitaxel (175 mg/m^2^) plus carboplatin (AUC 6) with either placebo (control) or ipilimumab 10 mg/kg administered concurrently with chemotherapy (concurrent ipilimumab) or after two doses of chemotherapy (phased ipilimumab). The rational for the phased regimen was to induce an antigen release with chemotherapy before ipilimumab exposure, with the aim to increase the effect of immunotherapy. Treatment was administered in the induction phase every three weeks for 18 weeks, followed by maintenance ipilimumab or placebo every 12 weeks. The phased-ipilimumab regimen improved immune related (ir) PFS compared with control (median irPFS 6.4 versus 5.3 months; HR: 0.64; *p* = 0.03), while the concurrent ipilimumab regimen did not (HR, 0.75; *p* = 0.11). On the basis of the promising results of this trial, a randomized, double blind phase III trial evaluated the efficacy and the safety of ipilimumab or placebo plus platinum-etoposide in patients with ES-SCLC [17]. Eligible patients were randomized to receive ipilimumab 10 mg/kg or placebo in a phased dosing regimen. During the first four cycles of induction, patients in both arms received etoposide 100 mg/m^2^ on days 1–3 plus investigator’s choice of platinum agent on day one of each cycle (cisplatin 75 mg/m^2^ or carboplatin AUC 5). During cycles three and four, patients were treated with ipilimumab 10 mg/kg or placebo every three weeks in addition to chemotherapy; during cycles five and six, patients were treated only with ipilimumab or placebo. Overall, 1132 patients were randomized in the study: 40% were ≥65 years old, 70% had an ECOG PS of 1 and 11% had brain metastases. No advantage was found in terms of OS in the experimental group, with a median OS of 11 versus 10.9 months (HR: 0.94; 95% CI, 0.81 to 1.09; *p* = 0.3775). Median PFS was 4.6 versus 4.4 months for control and experimental arm, respectively (HR: 0.85; 95% CI: 0.75 to 0.97). The combination of ipilimumab and chemotherapy was associated with a numerically higher frequency of any grade treatment-related AEs (27% vs. 13%), serious treatment-related AEs (22% vs. 11%), including diarrhea, rash, and colitis and a higher rate of treatment-related discontinuation (18% versus 2%). Five treatment-related deaths occurred with chemotherapy plus ipilimumab and two with chemotherapy plus placebo.

## 4. Checkpoint Inhibitors as Maintenance Therapy

The efficacy of a maintenance strategy with checkpoint inhibitors has been explored with anti CTLA-4 and anti-PD-1 inhibitors. In a phase II study the anti PD-1 pembrolizumab was administered at 200 mg/m^2^ within 8 weeks of the last cycle of platinum-etoposide chemotherapy, as maintenance therapy for up to two years [19]. Forty-five patients were enrolled in the study: median age was 66 years, 56% were female and 22% had brain metastases. Median PFS was 1.4 months (95%CI: 1.3–2.8) and median OS was 9.6 months (95%CI: 7.0–12): these data suggest the lack of activity of pembrolizumab as maintenance therapy. The 1-year PFS and OS rate were 13% and 37%, suggesting that a subgroup of patients could derive a clinical benefit. Tumor samples were collected for PDL-1 assessment in tumor cells and in stromal tissue: three patients had PDL-1 expression on tumor cells and eight were positive at stromal interface. No differences were found for patients with PDL-1 positive on tumor cells, but a longer PFS and OS was found for those with high PDL-1 expression in stromal tissue. In the CheckMate 451, a double-blind phase III study, 834 patients who did not progressed after four cycles of platinum-based chemotherapy, were randomized to receive a combination immunotherapy with nivolumab and ipilimumab, nivolumab alone, or placebo [20]. Patients received nivolumab 1 mg/kg plus ipilimumab 3 mg/kg every 3 weeks i.v for 4 doses followed by nivolumab 240 mg every two weeks or nivolumab 240 mg every two weeks, or placebo for up to 2 years. The study did not meet its primary endpoint: overall survival was not prolonged with nivolumab plus ipilimumab versus placebo (HR: 0.92; 95% CI 0.75–1.12; *p* = 0.3693). Overall survival with nivolumab versus placebo (secondary endpoint) was also not prolonged (HR: 0.84; 95% CI 0.69–1.02). A modest improvement in PFS was observed with ipilimumab plus nivolumab (HR: 0.72; 95%CI: 0.60–0.87) and nivolumab (HR: 0.67, 95%CI: 0.56–0.81) versus placebo.

## 5. Checkpoint Inhibitors in Pretreated Patients

Checkpoint inhibitors as single agents showed promising anti-tumor activity and good safety profile in patients with SCLC after platinum-based therapy and at least one other prior line of therapy (Table 2). The efficacy of pembrolizumab has been evaluated in two trials: Keynote 028 and Keynote 158. Keynote 028 was a phase Ib trial that investigated the tolerability and activity of pembrolizumab in twenty different tumor types; SCLC patients were enrolled in cohort C1 [21]. Patients were eligible if the tumour had progressed after standard treatment or if standard treatment was not appropriate for the patient. Moreover, tumour progression had to be histologically or cytologically confirmed and PDL-1 positivity was also required. Patients with active brain metastases or who were treated with radiotherapy for brain metastases in a period shorter than 12 weeks were excluded. Primary end point was overall response rate. Twenty-four patients were eligible and they were treated with pembrolizumab 10 mg/kg every two weeks for 24 months or until disease progression. Overall, most of patients (87.5%) were treated with two or more lines of prior therapy, 37.5% with three or more. The ORR was 33.3%; median duration of response was 19.4 months (range, ≥3.6 to ≥20.0 months). Median PFS was 1.9 months (95%CI: 1.7 to 5.9 months). Median OS was 9.7 months (95%CI: 4.1 months to not reached); the 6- and 12-month OS rates were 66.0% and 37.7%, respectively. Treatment related adverse events were seen in 16 of 24 patients; the most frequent were arthralgia, asthenia, and rash; one patient experienced grade 5 colitis/intestinal ischemia, which was considered as only probably related to pembrolizumab. The phase II Keynote 158 study investigated the activity of pembrolizumab in 11 different types of solid cancer: the cohort G was dedicated to SCLC patients [22]. Patients were eligible if they had tumour progression after standard treatment or if standard treatment was not appropriate, a life expectancy of more than three months and if there had evaluable tumour samples for PDL-1, although PDL-1 positivity was not requested for the trial. Primary end point was ORR. All patients were treated with pembrolizumab 200 mg every three weeks. Overall, 107 patients were eligible: ORR was 18.7% (95%CI: 11.8–27.4); median PFS was of 2 months (95%CI: 1.9–2.1) and, notably, it was similar in both PDL1 positive and negative patients (2.1 and 1.9 months, respectively). Median OS was 9.1 months (95% CI, 5.7–14.6): 14.6 months in patients with PD-L1–positive tumours and 7.7 months in patients with PD-L1–negative tumours. Adverse events were reported in 59% of patients, leading in 4 cases to discontinuation of treatment and in 1 case to death due to pneumonia. Recently, a pooled analysis of the aforementioned two studies was published, reporting updated data from both trials [23]. Primary end point of the analysis was ORR, secondary end points were PFS, OS and safety. Only patients who had previously received two lines of chemotherapy were considered for the analysis: therefore, 83 patients of the original 131 patients from both trials were included (19 patients from Keynote 028 and 64 patients from Keynote 158). ORR was 19.3% (95% CI: 11.4–29.4); of note, 88% of responders were PDL-1 positive. The median PFS was 2.0 months (95%CI: 1.9–3.4), while the median OS was 7.7 months (95% CI: 5.2–10.1). Adverse events were observed in 61.4% of patients, but only 9.6% had grade 3–5 adverse events. The most common events were fatigue, pruritus and cutaneous rash; two grade 5 events were recorded, including an intestinal ischemia and a pneumonitis. On June 2019, the Food and Drug Administration granted accelerated approval to pembrolizumab for metastatic SCLC patients with disease progression after platinum-based chemotherapy and at least one other prior line of therapy.

The activity of nivolumab in pretreated patients with SCLC has been evaluated in the Checkmate 032 and Checkmate 331 clinical trials. Checkmate 032 was a phase I,II basket trial that tested nivolumab alone or nivolumab plus ipilimumab in different tumors, including metastatic SCLC [24]. In the SCLC cohort, patients with limited and extensive stage disease who had progressed after at least one line of chemotherapy with cisplatin were included. Patients were non-randomized, but sequentially added to each different group. Patients received nivolumab 3 mg/kg every two weeks until progression or unacceptable toxicity, or nivolumab plus ipilimumab every three weeks for four cycles, then only nivolumab every three weeks until progression or unacceptable toxicity. In the nivolumab plus ipilimumab arm patients were initially treated with nivolumab 1 mg/kg plus ipilimumab 1 mg/kg, in order to assess the tolerability of the combination. Then they were included in the two main arms: nivolumab 3 mg/kg plus ipilimumab 1 mg/kg or nivolumab 1 mg/kg plus ipilimumab 3 mg/kg. Nine patients after progression from nivolumab alone group crossed over to nivolumab plus ipilimumab combination groups: one patient in nivolumab 1 mg/kg plus ipilimumab 3 mg/kg and eight patients in nivolumab 3 mg/kg plus ipilimumab 1 mg/kg. Overall, 216 patients were enrolled: 98 in nivolumab 3 mg/kg arm, three in nivolumab 1 mg/kg plus ipilimumab 1 mg/kg arm, 61 with nivolumab 1 mg/kg plus ipilimumab 3 mg/kg arm and 54 in nivolumab 3 mg/kg plus ipilimumab 1 mg/kg arm. An objective response was observed in 10% of patients treated with nivolumab alone, in 23% of patients treated with nivolumab 1 mg/kg plus ipilimumab 3 mg/kg, and in 19% of patients treated with nivolumab 3 mg/kg plus ipilimumab 1 mg/kg. Median OS was 4.4 months in the nivolumab alone group, 7.7 months in the nivolumab 1 mg/kg plus ipilimumab 3 mg/kg group and 6 months in the nivolumab 3 mg/kg plus ipilimumab 1 mg/kg group. Median PFS was 1.4 months in the nivolumab alone group, 2.6 months in the nivolumab 1 mg/kg plus ipilimumab 3 mg/kg group and 1.4 months in the nivolumab 3 mg/kg plus ipilimumab 1 mg/kg group. The response rate was not related to PDL-1 status. In the nivolumab alone group, 53% of patients experienced adverse events of any grade, 13% being G3–4; no grade 5 treatment-related events were reported; the most frequent ones in this group were fatigue, pruritus, diarrhoea; 6% of patients had to discontinue the treatment due to adverse events. In the nivolumab 1 mg/kg plus ipilimumab 3 mg/kg group 82% of patients experienced adverse events of any grade, 38% being G3–G5; 2 treatment related deaths were reported, due to myasthenia gravis and worsening of kidney failure; the most frequent events in this group were fatigue, pruritus and diarrhoea; 11% of patients had to discontinue treatment due to adverse events. In the nivolumab 3 mg/kg plus ipilimumab 1 mg/kg group, 76% of patients reported adverse events of any grade, 20% of them being G3–5; one grade 5 treatment related event was reported, due to pneumonitis; the most frequent adverse events were fatigue, diarrhoea and decreased appetite; 6% of patients had to discontinue treatment due to adverse events. An updated analysis was presented after a follow up of 18 months and it showed ORR in 11% of patients treated with nivolumab monotherapy and in 25% of patients treated with nivolumab 1 mg/kg plus ipilimumab 3 mg/kg [31]. A median OS of 4.1 months and 7.9 months was observed in the nivolumab monotherapy and nivolumab 1 mg/kg ipilimumab 3 mg/kg group, respectively. Adverse events of any grade were reported in 60% of patients in the nivolumab alone group and in 82% of patients in the nivolumab plus ipilimumab group. These positive early results prompted an expansion randomized cohort of the study: in this expansion cohort, patients were randomized in two arms in a 3:2 fashion to nivolumab 3 mg/kg vs. nivolumab 1 mg/kg plus ipilimumab 3 mg/kg. A pooled analysis including only patients treated with nivolumab (59 from the non-randomized cohort and 50 from the randomized cohort of the study) and who had received at least two previous treatments was recently presented [25]. ORR was 11.9%; median DOR was 17.9 months. Median PFS was 1.4 months, while median OS was 5.6 months. Duration of responses was longer than 6 and 12 months in 77% and 62% of patients, respectively. PD-L1 tumor status was not predictive of response. Adverse event were reported in 55% of patients, while 11.9% of patients experienced G3–4 events. One G5 event due to pneumonitis was reported. Following these results, on August 2018 the Food and Drug Administration granted accelerated approval to nivolumab for SCLC patients with disease progression after platinum-based chemotherapy and at least one other line of therapy. Results from the expansion cohort of randomized only patients were recently published: 147 patients were randomized to receive nivolumab 3 mg/kg, while 96 were randomized to nivolumab 1 mg/kg plus ipilimumab 3 mg/kg for four cycles, continuing thereafter only nivolumab 3 mg/kg [26]. Patients were stratified for number of previous treatments (only one or at least two). ORR was 11.6% in the nivolumab group and 21.9% in the combination group. Median PFS was 1.4 months in nivolumab group and 1.5 months in the combination group, while median OS was 5.7 months in the nivolumab group and 4.7 months in the combination group. Adverse events were observed in 53.7% of patients in the nivolumab only group and 68.8% of patients in nivolumab plus ipilimumab group. Grade 3–4 events occurred in 12.9% of patients in the nivolumab alone group and in 37.5% of nivolumab + ipilimumab group. The most frequent events were fatigue, pruritus and arthralgia in the nivolumab group, while they were diarrhea, fatigue and pruritus in the nivolumab plus ipilimumab group. In the nivolumab alone group there was one death due to toxicity, while there were three deaths in the nivolumab plus ipilimumab group, caused by autoimmune related hepatitis, pneumonitis and encephalitis; one more death in the combo group was caused both by autoimmune colitis and disease progression. Checkmate 331 was a phase 3 trial that compared nivolumab (240 mg i.v every 2 weeks) versus chemotherapy (topotecan at 1.5 mg/m^2^ iv or 2.3 mg/m^2^ oral daily on days 1–5 or amrubicin 40 mg/m^2^, iv daily on days 1–3, where available) in pretreated patients with one line of cisplatin based chemotherapy [27]. Patients were stratified as platinum responders and non-platinum responders. Primary outcome measure was overall survival. The trial didn’t meet the endpoint, because median OS was 7.5 months in nivolumab versus 8.4 months in chemotherapy arm (HR: 0.86; CI: 95%: 0.72–1.04). However, curves showed a delayed separation after month 12 and, more interestingly, HR for OS of nivolumab versus chemotherapy in patients with platinum resistant SCLC was 0.71 (95% CI: 0.54–0.94). All grade adverse events occurred in 55% of patients treated with nivolumab and in 90% of patients treated with chemotherapy.

The activity of durvalumab has been evaluated in a phase I/II study in 21 patients with pretreated ES-SCLC. Durvalumab was administered at the dose of 10 mg/kg every two weeks for up to one year [28]. Median PFS was 1.5 months (95%CI: 0.9–1.8), median OS was 4.8 months (95%CI: 1.3–10.4), and 12-month OS rate was 27.6% (95%CI: 10.2–48.4). There were no drug discontinuations due to related adverse events or death and the safety profile was as expected. The activity of durvalumab (1500 mg) plus tremelimumab (75 mg) every four weeks for up to four months, followed by durvalumab monotherapy from week 16 until PD or discontinuation in patients with platinum-refractory/resistant ED-SCLC was assessed by the Arm A of the phase II BALTIC study (NCT02937818). Preliminary results from this study showed that the response rate was 9.5% (95%CI: 1.17–30.38). In addition five patients had stable disease (23.8%) and one patient had an unconfirmed partial response [29]. Grade 3 or higher adverse events were reported in 10 patients (48%).

Atezolizumab as single agent failed to show significant activity in the randomized phase II IFCT-1603 study in 73 patients with ES-SCLC after failure of first line platinum etoposide-basing chemotherapy [30]. Patients were randomized to receive either atezolizumab (1200 mg intravenously every three weeks) or second line treatment with topotecan (up to six cycles) or re-induction of initial chemotherapy. After a median follow-up of 13.7 months, the median PFS was 1.4 versus 4.3 months in the atezolizumab and chemotherapy group, respectively. Moreover, no difference was found between the two arms for OS. There were no treatment-discontinuation or deaths related to immune-toxicities in the experimental arm.

## 6. Other Immunotherapy Approaches

Other immunotherapy strategies are being recently explored, including chimeric antigen receptor (CAR) T cells, tumor vaccines, antibody-drug conjugates (ADCs), immunomodulators. Some cell surface molecules such as CD56, delta-like protein 3 (DLL3), and CD47 are highly expressed in SCLC and they are potential targets of CART therapy. Preclinical studies showed that CD56 CAR + T cells were capable of killing in vitro CD56+ SCLC tumor cells and when tested against CD56 + human xenograft SCLC models, they were able to inhibit tumor growth in vivo [32]. These results indicate that CD56-CARs merit further investigation as a potential treatment for CD56 + malignancies. AMG 119 is an adoptive cellular therapy, consisting of a patient’s own T cells genetically modified ex vivo to express a chimeric antigen receptor (CAR) that targets DLL3 and redirects cytotoxic T cells to DLL3-positive cells. AMG 119 showed potent eradication of DLL3-positive cells in vitro and a single administration of AMG 119 in vivo reduced mean tumor volume in a SHP-77 xenograft model [33]. AMG 119 is currently under investigation in a phase I study in patients with advanced SCLC in progression after at least one platinum-based regimen (clinicaltrial.gov: NCT03392064). A similar strategy using the bispecific T-cell engager (BiTE) AMG 757 targeting DLL3 is also ongoing (clinicaltrials.gov: NCT03319940). CD47 plays a role in blocking phagocytosis, improving tumor survival, metastasis, and angiogenesis: CD47-CAR-T cells are expected to kill cells with high expression of CD47 and to be a potential treatment for SCLC [34,35]. Tumor vaccines can bind to tumor associate antigens (TAAs) to induce cellular and humoral immune responses, stimulating dendritic cells (DC) and activating CD8 T cell. Although several TAAs have been found in SCLC cells including the fucosylGM1, ganglioside GD3, polysialic-acid, P53 protein, most clinical trials on tumor vaccines have failed to improve the OS of patients [36]. ADCs can kill tumor cells by identifying TAAs in tumor cells. An antibody-drug conjugate, rovalpituzumab tesirine (Rova-T) was designed to specifically target delta-like ligand 3 (DLL3), an inhibitory Notch ligand that is highly expressed in SCLC (80%) and other neuroendocrine tumors but minimally expressed in normal tissues [37]. However, after initial promising signal of activity, a phase III study comparing Rova-T versus topotecan in the second line in patients with advanced SCLC (TAHOE; clinicaltrials.gov: NCT03061812) was discontinued after the results of an interim analysis showing that Rova-T was not superior to topotecan. A phase III study MERU (clinicaltrials.gov: NCT03033511) is currently ongoing to evaluate the efficacy of Rova-T for patients with SCLC as maintenance therapy following first-line chemotherapy. Finally, trials using a number of immunomodulators, including interleukin-2 (IL-2), interferons (IFNs) were also conducted in SCLC with negative results. More recently lefitolimod (MGN1703), an agonist of toll-like receptor 9 (TLR9), demonstrated favorable tolerability and immune activity in early studies [38]. TLR9 is expressed on a variety of immune cells and plays a major role in activation of innate immunity including stimulation of cytokine production, including type 1 IFNs. In the phase II IMPULSE study, lefitolimod was evaluated as maintenance treatment in patients with ES-SCLC after objective response to first-line chemotherapy. Although no relevant effect of lefitolimod was observed on the main efficacy end point in the ITT population, a subgroup analysis showed a reduction in the risk of death in patients with low frequency of activated CD86+ B cells (HR: 0.53; 95%CI: 0.26–1.08) and in patients with chronic obstructive pulmonary disease (HR: 0.48; 95%CI: 0.20–1.17) [39].

## 7. Discussion

Immune checkpoint inhibitors are the first agents in the last decades to determine an improvement in outcomes of ES-SCLC patients. In the IMpower 133 and CASPIAN studies, the addition of atezolizumab and durvalumab, respectively, to first-line chemotherapy in previously untreated patients with ES-SCLC produced a significant improvement in OS with an acceptable safety profile, leading to a new standard of care for the first-line therapy of patients with ES-SCLC. On the contrary, no benefit was observed with checkpoint inhibitors as maintenance or second-line therapy, while nivolumab and pembrolizumab single agents were approved in patients with SCLC after at least one other prior line of therapy. However, although ICIs in combination with standard chemotherapy improved OS in ES-SCLC, the results observed so far with immunotherapy cannot be considered the awaited breakthrough in SCLC, due to the small improvement in overall survival, the benefit limited to a small number of patients, and the lack of predictive biomarkers.

A number of relevant questions remain to be addressed: why did pembrolizumab fail to improve OS in KEYNOTE-604 study? What is the best PDL-1 inhibitor to combine with chemotherapy in the first-line setting? What will be the role of predictive factors? What will be the role of PDL-1 inhibitors in patients with LS-SCLC and in combination with radiotherapy? How the SCLC treatment scenario is evolving?

For the first question, although in the KEYNOTE-604 study OS was not significantly prolonged with the addition of pembrolizumab to chemotherapy, durable responses in a subset of patients and a statistically significant advantage in PFS were observed in the experimental arm. A possible explanation for the negative impact of the addition of pembrolizumab on OS could be the inclusion in KEYNOTE-604 study of patients with poorer prognostic factors, suggested by a worse OS reported in the study. In particular, KEYNOTE-604 included more patients older than 65 years, with brain metastases, an ECOG PS of 1, larger tumors, elevated LDH concentrations and ≥3 sites of metastases than IMpower 133 and CASPIAN studies. Moreover, in KEYNOTE-604 an imbalance in terms of number patients randomized to experimental and control arm (14.5% vs. 9.8%) was observed among those with brain metastases, the only subgroup that did not seem to benefit from pembrolizumab plus chemotherapy. Therefore, these differences in baseline characteristics of patients among the three studies could explain a different impact of ICIs on overall survival.

For the second question, no direct comparison exists to date between atezolizumab and durvalumab and the magnitude of the benefit appears quite similar in IMpower 133 and CASPIAN trial, despite some differences between the two studies, including the number of cycles of chemotherapy in the standard arm (up to six cycles) and the use of carboplatin or cisplatin at investigator’s choice in the CASPIAN study. Additional information on the safety of the combination of atezolizumab plus carboplatin and etoposide for ES-SCLC patients in a real world population will come from the phase IIIb Mauris trial (clinicaltrials.gov: NCT04028050) (Table 3). Primary end points are incidence of serious adverse events and incidence of serious and non-serious immuno-mediated adverse events; results are expected by January 2023. The Oriental trial (clinicaltrials.gov: NCT04449861) is a phase IIIb trial investigating the safety of durvalumab plus standard chemotherapy in ES-SCLC Chinese patients. All patients will receive durvalumab plus cisplatin or carboplatin and etoposide. Primary end point is incidence of adverse events major or equal to grade 3 per CTCAE. Results are due in January 2023. The role of nivolumab in combination with platinum and etoposide vs. placebo plus platinum and etoposide in the first-line therapy of ES-SCLC will be evaluated by a phase II study, the EA5161 trial (clinicaltrial.gov: NCT03382561). The primary objective is PFS, while secondary objectives are OS, ORR and adverse events. Results are awaited in June 2021.

For the third question, biomarkers to identify SCLC patients responding to ICIs are lacking. In contrast to the 60% of NSCLC which express the programmed death ligand 1 (PD-L1), only 18–32% of SCLC cases are PD-L1 positive [40]. Furthermore, level of PD-L1 expression in SCLC is not clearly associated with response to immunotherapy and the use of PD-L1 as predictive biomarker for immunotherapy is not recommended. Some studies assessed tumor mutational burden (TMB) by next generation sequencing (NGS) or whole exome sequencing (WES) in SCLC patients to test potential correlation with response to ICIs. Generally, a high TMB was associated with better PFS and OS, while ORR was not improved. PFS and OS tend to proportionally improve with increasing mutational load [41]. An exploratory analysis of the CheckMate 032 study found that 1y-PFS rates were higher in the high TMB group (21.2 and 30.0% for nivolumab monotherapy and nivolumab plus ipilimumab, respectively) compared with the low (not calculable and 6.2%, respectively) or medium (3.1 and 8.0%, respectively) TMB groups.

Similarly, 1y-OS rate was higher in the high TMB group (35.2 and 62.4% for nivolumab monotherapy and nivolumab plus ipilimumab, respectively) than in the low (22.1 and 23.4%, respectively) or medium (26.0 and 19.6%, respectively) TMB groups [42]. By contrast, exploratory subgroup analyses of the IMpower 133 and KEYNOTE-604 studies showed no clear correlation between TMB and clinical outcome in patients receiving chemotherapy plus ICIs. A novel blood-based TMB (bTMB) assay has been recently validated through a retrospective analysis of two randomized clinical trials from more than 1000 patients with advanced NSCLC treated with atezolizumab as second-line treatment [43]. This analysis demonstrated that TMB can be accurately measured in plasma and that bTMB is associated with clinical benefit from ICI therapy. These findings are important for future application of bTMB in clinical selection of lung cancer patients for ICI therapy and bTMB is currently under investigation in a randomized prospective phase III clinical trial (B-FAST study; NCT03178552).

In the IMpower 133 trial, the PD-L1 test was not performed due to the high rate of inadequate sample types, but evaluation of tumor mutational burden was performed by testing a blood-derived cell free DNA (cfDNA). An exploratory subgroup analysis showed that tumor mutational burden levels assessed on cfDNA at different cutoffs (10 or 16 mutations per megabase) did not predict benefit with atezolizumab [12]. The lack of a predictive effect of PDL-1 was observed in the CASPIAN study, where the HRs for PFS and OS were similar in patients with PD-L1–positive and PD-L1–negative tumors. The reason of the different behavior of SCLC compared with other tumors is not easily explainable. We could argue that SCLC is a very aggressive, proliferating and unstable tumor, hence TMB or different mutations driver of response in several other carcinomas including melanoma and NSCLC are less meaningful in SCLC. Other potential predictive factors could be considered, such as genomic changes involving tp53 and rb1. Moreover, expression of schlafen family member 11 (SLFN11), which promotes cell death following DNA damage, has been recently indicated as a biomarker of responsiveness in terms of PFS and OS in relapsed or refractory SCLC patients treated with temozolomide and the poly(ADP-ribose) polymerase (PARP) inhibitor veliparib [44]. The potential role of SLFN11 in the sensitization of SCLC to PARP-inhibitor is under investigation.

For the fourth question, immunotherapy is expected to have a synergistic effect when combined to radiotherapy, but final data are currently lacking, because consolidation thoracic radiation therapy was excluded in IMpower 133, CASPIAN and KEYNOTE-604 trials. Therefore, the role of thoracic radiotherapy remains a relevant issue in the current chemo-immunooncology era and a number of studies are evaluating the role of checkpoint inhibitors in combination with radiotherapy and/or in patients with LS-SCLC (Table 3). The Raptor trial (clinicaltrials.gov: NCT04402788) is a phase II/III study testing the efficacy of adding radiotherapy to atezolizumab in patients with ES-SCLC. All patients must have been treated with atezolizumab plus chemotherapy for 4–6 cycles, reporting a partial response or stable disease. Then, patients are randomized to receive atezolizumab maintenance alone or atezolizumab plus radiotherapy for five days a week for the first five weeks. Primary end point for phase 2 part of the trial is PFS; for the phase 3 part is OS. Final results are expected in May 2027 and should clarify the role of consolidative thoracic radiation in SCLC patients with extensive stage. The LU005 trial (clinicaltrials.gov: NCT03811002) is a phase II/III trial assessing the efficacy of atezolizumab in combination with radiotherapy in patients with LS-SCLC. Patients are randomized to receive one year of treatment with atezolizumab or placebo plus standard platinum plus etoposide for three cycles and concomitant radiotherapy. The primary end-points are PFS and OS. Results are awaited in January 2027 and should definitively answer the question on the efficacy of a combination strategy including immunotherapy in addition to chemotherapy and radiotherapy in SCLC patients with limited disease. Other clinical trials are evaluating the role of checkpoint inhibitors after chemoradiation. The ACHILES trial (clinicaltrials.gov: NCT03540420) is a phase II study for patients with LS-SCLC already treated with concurrent chemoradiation. Patients are randomized to receive atezolizumab for 1 year every three weeks or observation. Notably, also patients with an ECOG PS of 2 can enter the trial. The primary outcome is two year survival. Results are expected in January 2027. The ADRIATIC trial (clinicaltrials.gov: NCT03703297) is a phase III study for patients with LS-SCLC who have already been treated with concurrent chemoradiotherapy. In this trial patients are randomized to receive durvalumab plus placebo, durvalumab plus tremelimumab or placebo only. Primary outcome measures are OS and PFS. Results are awaited for January 2025 and should define also the role of a CTLA-4 inhibitor in addition to a PDL-1 inhibitor in SCLC patients with limited stage after chemoradiotherapy. The STIMULI trial (clinicaltrials.gov: NCT02046733) is another phase II study for patients with LS-SCLC already treated with concurrent chemoradiation. In this trial, patients are randomized to receive ipilimumab plus nivolumab for 4 cycles, then only nivolumab for a total of 1 year of treatment, or observation only. The primary end points are OS and PFS. Results are awaited for January 2022. Finally, a number of trials are evaluating new checkpoint inhibitors in patients with SCLC. The Skyscraper 02 (clinicaltrials.gov: NCT04256421) is a phase III trial that will investigate the role of tiragolumab, a novel cancer immunotherapy designed to bind to TIGIT, an immune checkpoint protein expressed on immune cells. Only naive ES-SCLC patients can enter the trial: patients are randomized to receive tiragolumab plus atezolizumab and carboplatin and etoposide or placebo plus atezolizumab and carboplatin and etoposide. Primary end points are PFS and OS. Results are awaited for October 2024. A phase 3 trial (clinicaltrials.gov: NCT03711305) is investigating the role of SHR-1316, a new immunoglobulin binding PD-L1 in Chinese naïve patients ES- SCLC. Patients are randomized to receive SHR-1316 plus carboplatin and etoposide or placebo plus carboplatin and etoposide. Primary end points are PFS and OS; results are awaited for January 2023. A phase 3 trial (clinicaltrials.gov: NCT04063163) is investigating the role of HLX10, another new PD-1 inhibitor. In this trial, patients are randomized in a 2:1 fashion to receive HLX-10 plus carboplatin and etoposide or placebo plus carboplatin and etoposide. This trial is enrolling naive patients with ES-SCLC. Primary endpoint is PFS. Results are awaited for January 2022.

For the fifth question, after decades of no progress, the treatment scenario of SCLC is now evolving. In the first line setting, atezolizumab and durvalumab combined with chemotherapy have shown to improve overall survival compared with chemotherapy alone, while in the second line setting, on 15 June 2020 the Food and Drug Administration granted accelerated approval to lurbinectedin for patients with metastatic SCLC in progression on or after platinum-based chemotherapy. Lurbinectedin, a selective inhibitor of oncogenic transcription, was administered at 3.2 mg/m^2^ as a 1-h intravenous infusion every 3 weeks in a phase II study including 105 patients with pretreated SCLC and it was active in terms of response rate (35.2%; 95%CI: 26–45%), with a median response duration of 5.3 months (95%CI: 4.1, 6.4) and acceptable safety profile [45]. The efficacy of lurbinectedin in combination with doxorubicin is currently being investigated as second line therapy in a randomised, phase 3 trial (ATLANTIS; clinicaltrials.gov: NCT02566993). A better understanding of the biology of SCLC and the recent identification of different molecular subtypes defined by differential expression of four key transcription regulators [46] may represent a significant step forward in defining a winning treatment strategy for SCLC.

## 8. Conclusions

In conclusion, the combination of atezolizumab or durvalumab with platinum-etoposide is a new standard of care for patients with ES-SCLC. Future challenges to optimize immunotherapy in SCLC are the identification of biomarkers predictive of response to checkpoint inhibitors in SCLC and the definition of the role of immunotherapy in patients with limited stage SCLC, in combination with radiotherapy or with other biological agents.

## Figures and Tables

**Table 1 cancers-12-02522-t001:** First line immunotherapy in ES-SCLC.

Author	Phase	No. Pts	Treatment	Response Rate (%)	Progression-Free Survival (Months)	Overall Survival (Months)
Horn L et al. [12] IMpower 133	III	403	Atezolizumab + carboplatinum-etoposide vs. placebo + carboplatinum + etoposide	60.2 vs. 64.4	5.2 vs. 4.3, HR: 0.77; 95% CI: 0.62–0.96, *p* = 0.02	12.3 vs. 10.3, HR: 0.70; 95%CI: 0.54–0.91, *p* = 0.007
Paz-Ares L et al. [14] CASPIAN	III	805	Durvalumab + platinum-etoposide vs. platinum-etoposide	79 vs. 70	5.1 vs. 5.4 HR: 0.78; 95%CI: 0.65–0.94	13.0 vs. 10.3, HR: 0.73; 95%CI: 0.59–0.91, *p* = 0.0047
Rudin CM et al. [16] KEYNOTE-604	III	453	Pembrolizumab + platinum-etoposide vs. placebo + platinum + etoposide	70.6 vs. 61.8	4.5 vs. 4.3, HR: 0.75; 95%CI: 0.61–0.91; *p* = 0.0023	10.8 vs. 9.7, HR: 0.80; 95%CI: 0.64–0.98; *p* = 0.0164
Reck M et al. [17] CA184–156 study	III	1132	Ipilimumab + platinum-etoposide vs. platinum-etoposide	62 vs. 62	4.6 vs. 4.4 HR: 0.85; 95%CI: 0.75–0.97	11.0 vs. 10.9 HR: 0.94; 95%CI: 0.81–1.09, *p* = 0.3775

Pts: patients; HR: hazard ratio.

**Table 2 cancers-12-02522-t002:** Immunotherapy in pretreated SCLC patients.

Author	Phase	Pts	Setting	Treatment	Response Rate (%)	PFS (Months)	OS (Months)
Ott et al. [21] Keynote 028	Phase Ib	24	PD after standard treatment; PDL 1+	Pembrolizumab (10 mg/kg)	33.3	1.9 (95%CI: 1.7–5.9)	9.7 (95%CI: 4.1–n.r.)
Chung et al. [22] Keynote 158	Phase II	107	PD after standard treatment	Pembrolizumab (200 mg)	18.7 (95%CI: 11.8–27.4)	2 (95%CI: 1.9–2.1)	9.1 (95%CI: 5.7–14.6)
Chung et al. [23] Keynote 028/158	Pooled analysis	83	III line	Pembrolizumab (10 mg/kg or 200 mg)	19.3 (95%CI: 11.4–29.4)	2.0 (95%CI: 1.9–3.4)	7.7(95%CI: 5.2–10.1)
Antonia et al. [24] Checkmate 032	Phase I,II	216	PD after at least one line with platinum based therapy	Nivolumab (3 mg/kg)	10 (CI95%: 5–18)	1.4 (CI95%: 1.4–1.9)	4.4 (CI95%: 3–9.3)
				Nivolumab (1 mg/kg) + Ipilimumab (3 mg/kg)	23 (CI95%: 13–36)	2.6 (CI95%: 1.4–4.1)	7.7 (CI95%: 3.6–18)
				Nivolumab (3 mg/kg) + Ipilimumab (1 mg/kg)	19 (CI95%: 9–31)	1.4 (CI95%: 1.3–2.2)	6 (CI95%: 3.6–11)
Ready et al. [25] Checkmate 032	Expansion cohort of phase II	109	III line	Nivolumab (3 mg/kg)	11.9 (CI95% 6.5–19.5)	1.4 (CI95% 1.3–1.6)	5.6 (CI95%: 3.1–6.8)
Ready et al. [26] Checkmate 032	Expansion cohort of randomized part	147	PD after at least one line with platinum based therapy	Nivolumab (3 mg/kg) vs. nivolumab (1 mg/kg) plus ipilimumab (3 mg/kg)	11.6 (95%CI: 6.9–17.9) vs. 21.9 (95%CI: 14.1–31.5) [odds ratio: 2.12; 95%CI: 1.06–4.26; p0.03]	1.4 (CI95%: 1.3–1.4) vs. 1.5 (95%CI: 1.4–2.2)	5.7 (CI95%: 3.8–7.6) vs. 4.7 (95%CI: 3.1–8.3)
Reck et al. [27] Checkmate 331	Phase 3	569	Second line	Nivolumab (240 mg) vs. topotecan 1.5 mg/m^2^ days 1–5 or amrubicin 40 mg/m^2^ days 1–3	14 vs. 16	1.4 (CI95%: 1.4–1.5) vs. 3.8 (CI95%: 3.0–4.2)	7.5 vs. 8.4 (HR 0.86 CI95%: 0.72–1.04)
Goldman JW et al. [28]	Phase I/II	21	Second/third line	Durvalumab 10 mg/kg	9.5% (95%CI: 1.2–30.4)	1.5 (95%CI: 0.9–1.8)	4.8 months (95%CI: 1.3–10.4)
Bondarenko I et al. [29] BALTIC, Arm A	Phase II	21	Second line	Durvalumab 1500 mg plus tremelimumab 75 mg	9.5% (95%CI: 1.1–30.3)	n.r.	n.r.
Pujol JL et al. [30]	Phase II	73	Second line	Atezolizumab 1200 mg vs. chemotherapy	2.3% (95%CI: 0.0–6.8)	1.4 (95%CI: 1.2–1.5) vs. 4.3 (95%CI: 1.5–5.9)	9.5 vs. 8.7 (HR: 0.84; 95%CI: 0.45–1.58; *p* = 0.60

Pts: patients; PD: progression disease; PFS: progression-free survival; OS: overall survival, n.r.: not reported.

**Table 3 cancers-12-02522-t003:** Ongoing trials with checkpoint inhibitors in SCLC.

Trial	Phase	Setting	Stage	Pts	Treatment	Primary End Points
NCT04028050 (MAURIS)	IIIb	First line	Extensive	150	Atezolizumab + carboplatin + etoposide	Incidence of SAE and of immuno-related events
NCT04449861 (ORIENTAL)	IIIb	First line	Extensive	300	Durvalumab + platinum + etoposide	Incidence of major events
NCT03382561	II	First line	Extensive	150	Nivolumab + platinum + etoposide vs. placebo + platinum + etoposide	PFS
NCT04402788 (RAPTOR)	II/III	Maintenance after first line therapy	Extensive	324	Atezolizumab + RT vs. atezolizumab alone	PFS (phase II) OS (phase III)
NCT03811002 (LU005)	II/III	Curative chemo-radiotherapy	Limited	506	Atezolizumab + chemo-radiotherapy vs. placebo + chemo-radiotherapy	PFS; OS
NCT03540420 (ACHILES)	II	Maintenance after curative chemo-radiotherapy	Limited	212	Atezolizumab vs. observation	2 year survival
NCT03703297 (ADRIATIC)	III	Maintenance after curative chemo-radiotherapy	Limited	600	Durvalumab + tremelimumab vs. durvalumab + placebo vs. placebo only	PFS; OS
NCT02046733 (STIMULI)	II	Maintenance after curative chemo-radiotherapy	Limited	264	NIvolumab + ipilimuamb vs. observation	OS; PFS
NCT04256421 (SKYSCRAPER 02)	III	First line	Extensive	400	Tiragolumab + atezolizumab + carboplatin + etoposide vs. atezolizumb + carboplatin + etoposide	PFS; OS
NCT03711305	III	First line	Extensive	396	SHR-1316 + carboplatin + etoposide vs. placebo + carboplatin + etoposide	PFS; OS
NCT04063163	III	First line	Extensive	489	HLX10 + chemotherapy vs. placebo + chemotherapy	PFS

Pts: patients; SAE: serious adverse event; PFS: progression-free survival; OS: overall survival.

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
