# Peer review of "Immunotherapy in Small Cell Lung Cancer"

_cancers, 2020, doi:10.3390/cancers12092522_

Round 1
Reviewer 1 Report
The authors reported the current scientific evidences on immune checkpoint inhibition in SCLC. The manuscript is well written and structured. Only minor spell check is required.
However, some considerations are needed.
1) Introduction section: an extensive description of results (ORR, mPFS, mOS) with topotecan in phase II/III trials should be reported.
2) PD-L1/PD-1 inhibitors in first line therapy:
- table 1: The reference refers to 4th studies reported in the table is 18 and not 17
3) CPIs in pretreated patients:
- The authors should describe the safety and activity/efficacy results from durvalumab trial [NCT01693562]
- Results from IFCT-1603 study with atezolizumab are lacking and they should be reported
- As combination CPIs, results from BALTIC study are lacking and should be discussed
4) A new section on the role of TMB (not explored in the manuscript) and PD-L1 expression as potential biomarkers of response to CPIs is suggested
5) Discussion section: although ICIs (durvalumab and atezolizumab) in combination with standard chemotherapy improved OS in ES-SCLC, do the authors consider so far immunotherapy the awaited breakthrough in SCLC? A critical point-of view by the authors is suggested
The authors give a possible explanation on negative results of KEYNOTE604 trial respect to those reported from IMpower133 and CASPIAN studies. I suggest authors to review extensively the included patient populations in IMpower 133 and CASPIAN studies (% of enrolled patients with brain metastases, % with liver involvement, % aged < 65 years, drop out/screening failure of 25%, etc). In this context, a more extensive and personal discussion on high/positive patients' selection (inclusion/exclusion criteria) in the latter studies should be explored. Are those the same patients we treat in clinical practice?
Author Response
Please see the attachment (Reviewer 1)

Reviewer 2 Report
This is a very thorough evaluation of the role of immunotherapy in SCLC. The authors do a good job of outlining prior and ongoing studies. There are some grammar issues that need addressing. Otherwise I have only a few comments that I feel would strengthen the manuscript.
Overall: The authors do a good job of providing background of SCLC, however I would suggest adding a paragraph dedicated to the molecular landscape of SCLC and why despite harboring biomarkers such as high TMB, overall the response to ICI in SCLC is more modest than for NSCLC or melanoma. Some discussion of genomic changes including tp53, rb1, and the recent identification of SLFN11 would be helpful.
Also, the discussion ends with essentially a list of ongoing trials. Would be best to leave most of that for the tables and highlight what each of the trials are answering.
Questions I would like to see added/addressed include (in addition to the molecular/pathology details as above):
- The role of consolidative thoracic radiation in the current era
- How does the approval of lurbinectedin fit in?
- At least a mention of novel immunotherapies such as chimeric antigen receptor T cell therapy and bispecific T cell engagers, several of which are being investigated in SCLC.
A few minor issues:
Abstract: "good safety profile" is too colloquial. Suggest "acceptable safety profile."
Section 2:
-Would specify rates of brain mets in IMpower 133 in each arm as you do for other studies since this is important prognostic indicator.
-"PDL1 test" is not precise: typically PD-L1 protein expression is assessed by IHC, either of tumor (TPS, tumor proportion score) or immune/tumor expression (composite score).
-Would specify that consolidation thoracic radiation (TRT) was excluded from IMpower133 and the role of TRT remains a question in the current chemo-IO era
-In discussing KN604, "...and it was associated with a benefit in OS not statistically significant" needs re-written. Likewise "rules of the protocol" is usually referred to as "protocol defined"
Section 3. Type, etoposide is D1-3 not 33. Also, there are two sections labeled 3.
Second Section 3: "higher PFS" doesn't make sense. PFS and OS would be longer or shorter. Could have higher rates of response, or higher rates of 2 yr survival.
Section 4: PDL1 positivity was required, not requested. It was eligibility requirement for this study.
Author Response
Please see the attachment (Reviewer 2)

Reviewer 3 Report
This is a comprehensive review of the evolving field of SCLC immunotherapy. While several drug combinations have been recently approved for the treatment of SCLC patients, there is still a need to pursue this type of basic and clinical research to improve the overall outcomes, which are still not desirable. With that, this article provides a valuable snapshot of the field, with a nice summary of past and future clinical trials. There is a brief introduction of the scientific background, but the main thrust of the article is devoted to clinical trials. It would be informative to include results of research devoted to the exploration of of other checkpoints in SCLC, besides the CTLA4, PD-1, PDL-1 that the article is focused on. This is merely a suggestion, not a "conditio sine qua non". There is a formatting issue towards the end of the article on p.11:unnecessary paragraphs occur at least in the PDF version downloaded by the reviewer. A couple of stylistic suggestions are also offered:
p.1 Recent evidence supports...
p.10 Results are expected in May 2027.
Author Response
Please see the attachment (Reviewer 3)

Round 2
Reviewer 1 Report
The authors reviewed the manuscript well. In my opinion, the manuscript is acceptable in this form.
Author Response
We thank the reviewer
Reviewer 2 Report
The authors have addressed all comments from the reviewers. The modified manuscript is improved after the additions and edits by the authors.
Author Response
We thank the reviewer.